

# Estimation of soil salt content by combining UAV-borne multispectral sensor and machine learning algorithms

Guangfei Wei[1,2], Yu Li[1], Zhitao Zhang[1,2], Yinwen Chen[3],
Junying Chen[1,2], Zhihua Yao[1,2], Congcong Lao[1,2] and Huifang Chen[1,2]

[1] College of Water Resources and Architectural Engineering, Northwest A&F University,
Yangling, China
[2] Key Laboratory of Agricultural Soil and Water Engineering in Arid and Semiarid Areas of
Ministry of Education, Northwest A&F University, Yangling, China
[3] Department of Foreign Languages, Northwest A&F University, Yangling, China

Corresponding authors
Yu Li, liyu2188@aliyun.com
Zhitao Zhang,
zhangzhitao@nwafu.edu.cn

## ABSTRACT

Soil salinization is a global problem closely related to the sustainable development of social economy. Compared with frequently-used satellite-borne sensors, unmanned aerial vehicles (UAVs) equipped with multispectral sensors provide an opportunity to monitor soil salinization with on-demand high spatial and temporal resolution. This study aims to quantitatively estimate soil salt content (SSC) using UAV-borne multispectral imagery, and explore the deep mining of multispectral data. For this purpose, a total of 60 soil samples (0–20 cm) were collected from Shahaoqu Irrigation Area in Inner Mongolia, China. Meanwhile, from the UAV sensor we obtained the multispectral data, based on which 22 spectral covariates (6 spectral bands and 16 spectral indices) were constructed. The sensitive spectral covariates were selected by means of gray relational analysis (GRA), successive projections algorithm (SPA) and variable importance in projection (VIP), and from these selected covariates estimation models were built using back propagation neural network (BPNN) regression, support vector regression (SVR) and random forest (RF) regression, respectively. The performance of the models was assessed by coefficient of determination ($R^2$), root mean squared error (RMSE) and ratio of performance to deviation (RPD). The results showed that the estimation accuracy of the models had been improved markedly using three variable selection methods, and VIP outperformed GRA and GRA outperformed SPA. However, the model accuracy with the three machine learning algorithms turned out to be significantly different: RF > SVR > BPNN. All the 12 SSC estimation models could be used to quantitatively estimate SSC (RPD > 1.4) while the VIP-RF model achieved the highest accuracy ($R_c^2 = 0.835$, $R_P^2 = 0.812$, RPD = 2.299). The result of this study proved that UAV-borne multispectral sensor is a feasible instrument for SSC estimation, and provided a reference for further similar research.

## INTRODUCTION

Soil salinization is a global ecological and environmental problem, which has become one of the main obstacles to agricultural production and sustainable economic development, especially in arid and semi-arid areas (*Wu et al., 2008*). The process of salinization is mainly due to the comprehensive influence of specific natural conditions (geological, topographic and climatic conditions) and improper agricultural practice. It is reported that salinized soil has a high concentration of salt ions, directly affecting the normal growth, yield and quality of crops (*Tavakkoli et al., 2011*; *Munns, 2010*). To manage and utilize saline soil, some effective methods are needed to acquire the saline soil information quickly, dynamically and accurately in arid and semi-arid areas.

Soil salt content (SSC) is an effective evaluation indicator of soil salinization (*Gorji, Tanik & Sertel, 2015*). Field-based (in-situ) are logistically difficult such as labor intensive and time-consuming (*Metternicht & Zinck, 2003*). Many scholars monitored soil salinization using satellite-borne remote sensing (RS) data together with field measurement over last two decades (*Yu et al., 2018*; *Allbed, Kumar & Sinha, 2014*; *Ivushkin et al., 2019b*), yet satellite images can be easily affected by bad weather and unfavorable revisit times. The recent development of unmanned aerial vehicle (UAV) has ushered in a new era enabling monitoring of environment and agriculture at unprecedented temporal and spatial, especially in the monitoring of such soil ingredients as moisture, heavy metals and organic carbon (*Gilliot, Vaudour & Joël, 2016*; *Bian et al., 2019*; *Yi et al., 2018*; *Easterday et al., 2019*; *Jay et al., 2018*). There have been a few cases involving the application of UAV-borne RS in soil salinization monitoring. For instance, *Ivushkin et al. (2019a)* combined three UAV-borne sensors to measure salt stress in quinoa plants, and their result showed UAV-borne RS was a useful technique to measure salt stress in plants. *Hu et al. (2019)* characterized and estimated soil salinity using a hyperspectral and electromagnetic induction (EMI) equipment mounted on a UAV platform, and concluded that UAV-borne hyperspectral imager was an effective tool for field-scale soil salinity monitoring and mapping. Aside from these studies, existing research showed that the majority of UAV-borne soil salinity estimations were conducted using hyperspectral or thermal cameras while there are few reports on the application of UAV-borne multispectral imagery to soil salt estimations. Compared with hyperspectral sensors, multispectral sensors have a much lower cost, and the band preprocessing is relatively simple, so it is significative to assess the capability of UAV-borne multispectral sensor in SSC estimation.

The spectral index, a composition of different spectral wavebands, has been frequently used to establish the correlation between spectral data and the information of soil salinization (*Allbed, Kumar & Sinha, 2014*; *Allbed, Kumar & Aldakheel, 2014*). A series of common salt indices, including Normalized Difference Salinity Index (NDSI) (*Zewdu, Suryabhagavan & Balakrishnan, 2017*), Salinity Index (SI) (*Allbed, Kumar & Aldakheel, 2014*) and Simple ratio index (SR) (*Chen, 1996*), have been widely used to represent soil salinization. Besides, some two-band (2D) indices and three-band indices (TBI) were proposed for the estimation of soil moisture content and soil electric conductivity, and the

results showed that these newly proposed indices containing more spectral information displayed a higher correlation with measured data (*Wang et al., 2019b*).

However, such indices construction will generate a large amount of redundant information, so it is important to filter the redundant information through variable selection method for model optimization. Research has shown that variable selection methods can improve the model predictive accuracy (*Hong et al., 2018*). Variable importance in projection (VIP) scores evaluates the importance of each variable in the projection used in a PLS model. Gray relational analysis (GRA) identify the primary and secondary relations among variables through the calculation and comparison of correlation sequence, and they both have been proved to be effective methods for variable selection (*Santos et al., 2019*; *Wang et al., 2018a*). Successive projections algorithm (SPA) is a variable selection method widely used in food and chemical engineering (*Diniz et al., 2015*; *Ghasemi-Varnamkhasti et al., 2012*; *Wu et al., 2009*). *Wang et al. (2019c)* proved the applicability of SPA method to soil salinity estimation, and the SPA method improved the accuracy of the inversion model. However, there are few studies on the application of these three variable selection methods to SSC estimation.

A common method to estimate SSC is the utilization of the mathematical statistical model using RS data, especially the linear regression model including partial least squares regression (PLSR) (*Farifteh et al., 2007*; *Xu & Wang, 2015*; *Sidike, Zhao & Wen, 2014*). But the relationship between spectral covariates and soil properties is rarely linear in nature (*Ge et al., 2019*). Machine learning provides an alternative mean of fitting nonlinear problems (*Nawar et al., 2016*). Machine learning algorithms, including back propagation neural network (BPNN), support vector regression (SVR), random forest (RF), extreme learning machine (ELM) and Cubist, have been extensively used in the quantitative estimation recently (*Hoa et al., 2019*; *Shataee et al., 2012*; *Maimaitiyiming et al., 2019*; *Li et al., 2015*; *Houborg & Mccabe, 2018*). The BPNN algorithm, strong in nonlinear fitting and self-learning, has been proven to be superior to traditional linear regression in prediction accuracy in many studies (*Pradhan & Lee, 2010*; *Bansal, Chen & Zhong, 2011*). The SVR algorithm grounded on kernel-based learning methods has the ability to solve nonlinear and high-dimensional problems. *Chen et al. (2015)* reported that SVR algorithm could effectively improve the accuracy of SSC estimation using the modified vegetation index. Many scholars have concluded that the RF algorithm has such unique advantages as small sample data processing, and nonlinear fitting problem solving (*Mutanga, Adam & Cho, 2012*; *Oliveira et al., 2012*). For instance, *Cushman & Wasserman (2018)* compared logistic regression and RF in multi-scale predictive model, and concluded the RF model had a much stronger ability to estimate presences and absences in the training set than logistic regression model. These studies showed that BPNN, SVR and RF are feasible methods for quantitative estimation. However, most of the existing researches focused on the comparison between machine learning algorithms and linear regression methods, and there is a lack of the combination of these three machine learning methods.

So far there are many researches on hyperspectral data mining (*Wang et al., 2019a*), but very few reports on deep mining of UAV-borne multispectral data. We try to answer the

following questions: The correlation between spectral indexes and SSC tends to vary, so which spectral indexes are suitable for this study and which variable selection method is the most effective? Which selection method and which regression algorithm can be combined to have the best prediction accuracy? These questions are unavoidable in UAV-borne multispectral monitoring of soil salinity.

Specifically, this study aimed to: (1) evaluate the potential and feasibility of UAV-borne multispectral RS for SSC quantitative estimation; (2) compare the effect of GRA, SPA and VIP on model accuracy, and investigate whether the three variable selection methods can improve the model predictive accuracy; (3) compare the prediction accuracy of BPNN, SVR and RF models, and identify the optimal SSC quantitative estimation model.

## MATERIALS AND METHODS

### Study area

Hetao Irrigation District (HID) is one of the three major irrigated areas in China. It is located in the upstream of the Yellow River and the west of Inner Mongolia. With a total irrigation area of 5740 km², HID is an important production base of cereal and oil plants in China. Shahaoqu Irrigation Area (107°05′~107°10′ E, 40°52′~41°00′ N), a typical region of saline soil in HID, was chosen as the study area (Fig. 1). Its terrain surface is relatively flat with the altitude from 1,034 m to 1,037 m. The climate is typically temperate continental, with its mean precipitation, evaporation and annual temperature of about 140 mm, 2,000 mm and 7 °C, respectively, and the frost-free period of 120–150 days. The local soil texture is mainly silty clay loam. Due to the unreasonable high irrigation, low drainage of local agriculture and the impact of such natural factors as geological, topographic and climatic conditions, the problem of salinization in this area is increasingly prominent, seriously affecting the development of local agricultural economy (*Wu et al., 2008*; *Gao et al., 2015*).

### Data collection

#### *Sample collection and chemical analysis*

The Hetao irrigation district administration gave field permit approval to us (No. 2017YFC0403302). Considering the various degree of salinization in Shahaoqu Irrigation Area, We chose four typical and representative areas (Area A, B, C and D, in Fig. 1) with different degrees of salinization in the farmland. Fifteen sampling cells (0.5 m × 0.5 m) of bare soil were distributed in each area (16 hm²). Altogether 60 sampling cells were identified, and the geographical position of each cell was recorded by GPS (Fig. 2). Samples were collected from May 14 to May 17, 2018, with a sampling depth of 0–20 cm. The soil samples were stored and sealed in aluminum boxes.

All the soil samples were dried, ground, and then passed through a 2.0 mm sieve to wipe off small stones and deadwood in the laboratory. Subsequently, the soil solution was prepared with the soil to water ratio of 1:5. After 8 h of full immersion, the electrical conductivity ($EC_{1:5}$, ds/cm) of the soil solution was measured by conductivity meter (DDS-307A; Shanghai Youke Instrument Branch, Shanghai, China), and then the SSC (%) was calculated by empirical formula: $SSC = (0.2882 \, EC_{1:5} + 0.0183)$ (*Huang et al., 2018*).

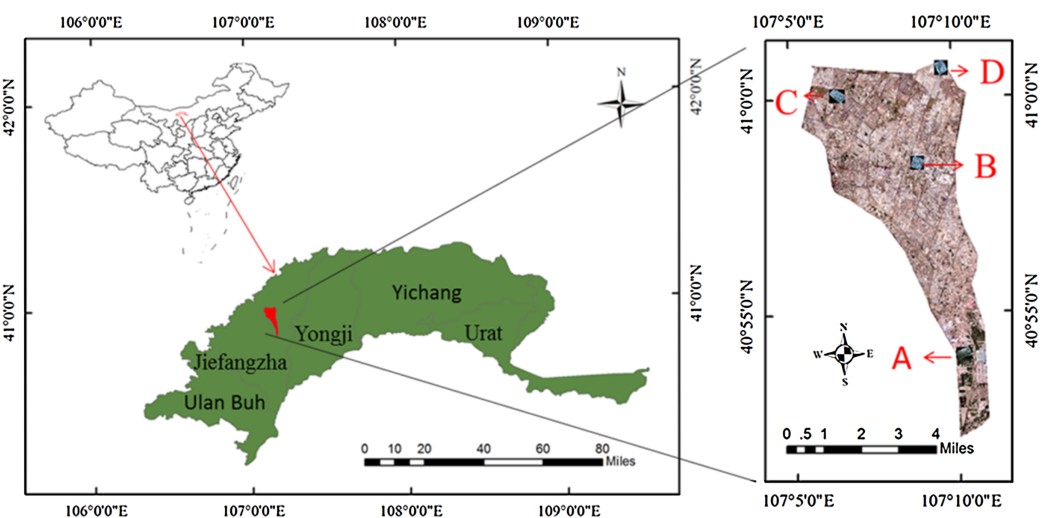

**Figure 1 The distribution of sampling point.** (A–D) The distribution of the four study areas respectively.

## Acquisition and processing of UAV-borne multispectral data

The flight platform adopted was the DJI Matrice 600 six-rotor UAV (Shenzhen Dajiang Innovation Technology Co., Ltd., Shenzhen, China) equipped with a Micro-MCA multispectral sensor (Tetracam Corporation, Chatsworth, CA, USA) (Fig. 3). The sensor is light, small and remote controllable. The parameters of the sensor are shown in Table 1. Simultaneous with the collection of soil samples, the UAV RS imagery of the four areas were acquired at 13:00 Beijing time every day from May 14 to May 17, 2018. The weather was sunny and windless, which is favorable for data acquisition. The UAV followed the fixed route at a height of 120 m, the spatial resolution of the imagery being 6.5 cm. The images were captured every 3 s and its overlap rate was 85%.

The post-processing of images included image mosaicking, geometric correction, radiation correction and orthographic correction with software Pix4D mapper 2.0.104 (Lausanne, Switzerland). Then four false color composite band images were generated and the sampling points of GPS information were input into the four images of the study area via software ENVI Classic. The gray values of the six bands of each sampling point and the whiteboard were extracted and the reflectance of the sampling points was obtained from the division of the former by the latter.

## Spectral indices construction

The spectral index is highly related to soil surface, which is an effective indicator to monitor soil salinization. Therefore, some of widely used soil salinity indices were chosen in this study. In addition, to find the sensitive spectral indices suitable for the study area and fully dig for the spectral data information, we built three 2D indices: the difference index (DI), the ratio index (RI) and the normalization index (NDI). The formulas of the spectral indices above are shown in Table 2.

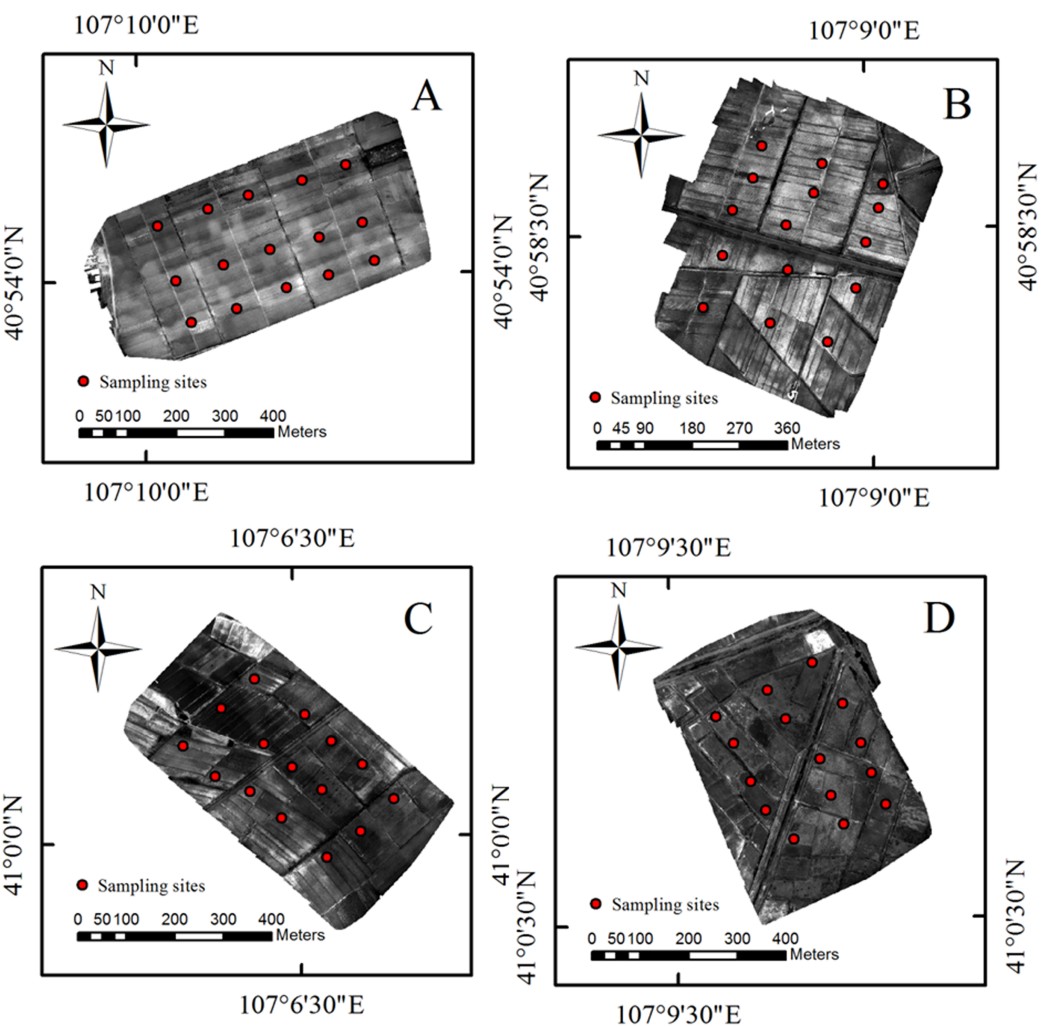

**Figure 2 The distribution of sampling point.** (A–D) The distribution of the four study areas respectively.

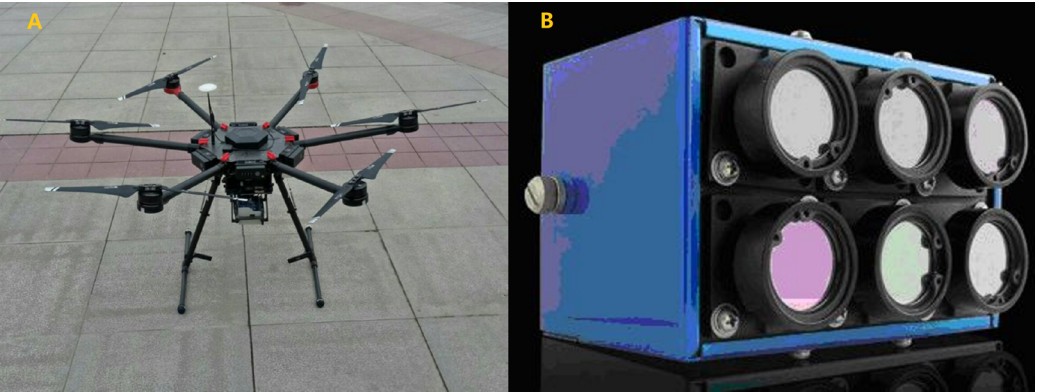

**Figure 3** (A) M600 unmanned aerial vehicle; (B) micro-MCA multispectral sensor.

**Table 1 MCA multispectral sensor parameters.**

| Parameter | Size |
| --- | --- |
| Weight/g | 670 |
| Field angle | 38.26 × 30.97 |
| The highest pixel | 1,280 × 1,024 |
| Band and band width/nm | B1 490 (10–25) |
| | B2 550 (10–25) |
| | B3 680 (10–25) |
| | B4 720 (10–25) |
| | B5 800 (10–25) |
| | B6 900 (10–25) |

**Table 2 Reference spectral indices.** B, G, R, NIR1 and NIR2 are spectral reflectance at wavelengths of 490 nm, 550 nm, 680 nm, 800 nm and 900 nm, respectively. $B_i$ and $B_j$ represent the reflectance values from random spectral bands available from the multispectral sensor.

| Spectral index | Formula | Full name | References |
| --- | --- | --- | --- |
| S1 | $B/R$ | Salinity Index 1 | *Allbed, Kumar & Aldakheel (2014)* |
| S2 | $(B - R)/(B + R)$ | Salinity Index 2 | |
| S3 | $(G \times R)/B$ | Salinity Index 3 | |
| S4 | $(B \times R)^{0.5}$ | Salinity Index 4 | |
| S5 | $(B \times R)/G$ | Salinity Index 5 | |
| S6-1 | $(R \times NIR1)/G$ | Salinity Index 6-1 | |
| S6-2 | $(R \times NIR2)/G$ | Salinity Index 6-2 | |
| SR-1 | $NIR1/R$ | Simple Ratio Index 1 | *Chen (1996)* |
| SR-2 | $NIR2/R$ | Simple Ratio Index 2 | |
| BI-1 | $(R^2 + NIR1^2)^{0.5}$ | Brightness Index 1 | *Khan et al. (2005)* |
| BI-2 | $(R^2 + NIR2^2)^{0.5}$ | Brightness Index 2 | |
| NDSI-1 | $(R - NIR1)/(R + NIR1)$ | Normalized Difference Salinity Index 1 | |
| NDSI-2 | $(R - NIR2)/(R + NIR2)$ | Normalized Difference Salinity Index 2 | |
| DI | $B_i - B_j$ | Difference Index | *Wang et al. (2019b)* |
| RI | $B_i/B_j$ | Ratio Index | |
| NDI | $(B_i - B_j)/(B_i \pm B_j)$ | Normalization Index | |

## Variable selection methods

### Gray relational analysis

GRA is an approach to measure the degree of correlation between factors according to the degree of similarity or difference in the development trend between factors. Its basic principle is to identify the primary and secondary relations among various factors through the calculation and comparison of correlation degrees (*Wang et al., 2018d*). GRA requires less data, and the principle is simple to grasp. The calculation steps of this method are: (1) determining comparison sequence and reference sequence; (2) normalizing the

spectral data; (3) calculating correlation coefficient; and (4) calculating correlation degree and ranking gray correlation degree (GCD). The calculation process is completed by software DPS 7.05 which is a data processing system made in China. The formula of GCD is:

$$GCD = \frac{1}{n} \sum_{i=1}^{n} \gamma(x_0(k), x_i(k)) \tag{1}$$

where $\gamma(x_0(k), x_i(k)) = \frac{\min_i \max_k |x_0(k) - x_i(k)| + \rho \min_i \max_k |x_0(k) - x_i(k)|}{|x_0(k) - x_i(k)| + \rho \min_i \max_k |x_0(k) - x_i(k)|}$, $\rho$ is the distinguishing coefficient with a value range [0,1]. In this study, $\rho$ was set as 0.5.

### Successive projections algorithm

SPA is a forward variable selection algorithm that minimizes collinearity in vector space, which uses a simple projection operation in vector space to obtain minimum co-linear variable subset. And it can greatly eliminate the co-linear effect between independent variables, thereby reducing the model complexity and improving the model stability and accuracy (*Araújo et al., 2001*). The core formula is as follows:

$$Px_j = x_j - \left(x_j^T x_{k(n-1)}\right) x_{k(n-1)} \left(x_j^T x_{k(n-1)}\right)^{-1} \tag{2}$$

where $P$ is the projection operator; all $j \in S$, $S$ represents the set of unselected variables; $K$ represents the selected independent variables.

### Variable importance in projection

VIP is a variable selection method based on PLSR which was first proposed by *Wold, Martens & Wold (1983)*. VIP scores offer a useful method to select the variables which contribute the most to the Y's variance explanation. And it is calculated as follows:

$$VIP = \sqrt{k \times \frac{\sum_{a=1}^{A} R_d(Y, t) W^2}{\sum_{a=1}^{A} R_d(Y, t)}} \tag{3}$$

where $K$ represents the total number of independent variables; $A$ is the number of components; $t$ represents the selected independent variables; $R_d(Y, t)$ represents the interpretation degree of components to dependent variables; $W^2$ represents the importance of variable in each component; the value of VIP > 1 indicates a strong relation between independent variables and dependent variables.

## Model construction and validation

In this study, three machine learning algorithms, including BPNN, SVR and RF, were applied to the quantitative inversion of SSC. The main method process is shown in Fig. 4. To ensure that calibration dataset and validation dataset can represent the statistical characteristics of the entire dataset, the division of samples was based on the Kennard–Stone (k–s) algorithm (*Kennard & Stone, 1969*), and 40 samples were selected as the calibration dataset and 20 samples as the validation dataset.

BPNN is a multi-layer feedforward neural network trained according to the error back propagation algorithm. It does not need to establish a specific mathematical model in data

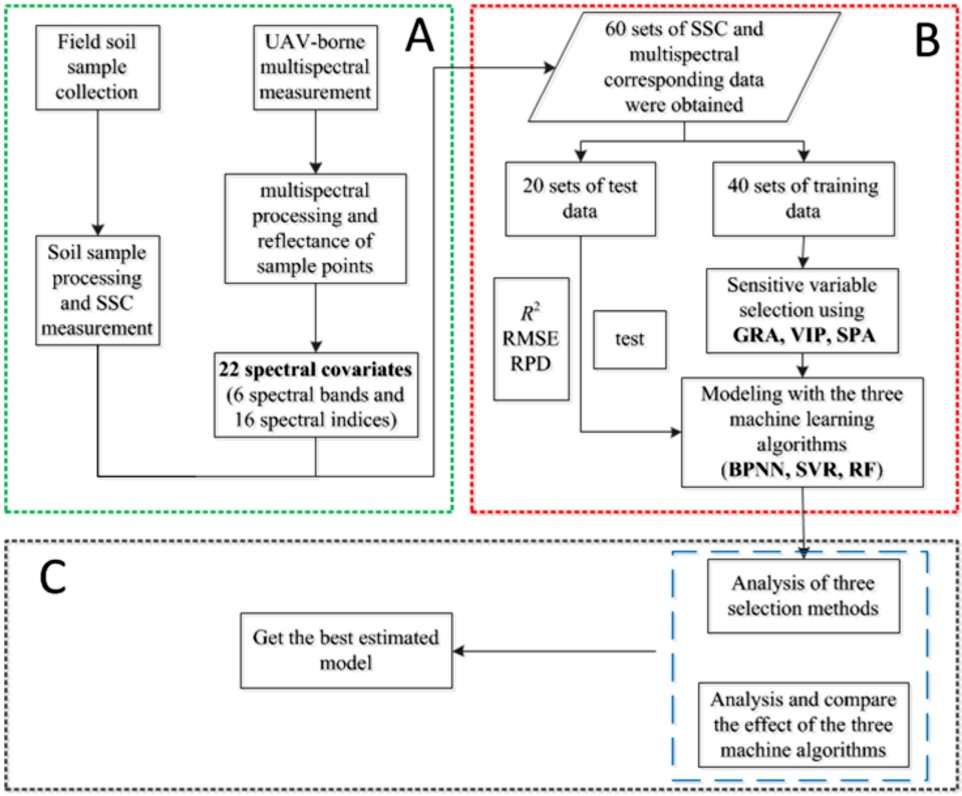

**Figure 4 Method flow chart.** (A) Data preprocessing; (B) modeling and (C) analysis.

analysis, and has strong fitting ability to figure out multi-factor nonlinear problems. The BPNN model topology includes the input layer, the hidden layer, and the output layer (*Wang et al., 2018c*). The sensitive variables selected by the three variable selection methods were input layers in this study, the measured SSC data were the output layer, and the size of hidden layer nodes was determined by cross-validation.

SVR is a machine learning algorithm based on the principle of structural risk minimization. It has the advantages of simple structure, strong adaptability, and powerful capability of tackling small sample, nonlinear and high-dimensional data problems (*Wang et al., 2019a*). In this study, the kernel function was set as polynomial kernel function (Polynomial). The training set cross-validation and grid search were used to optimize the parameters. The penalty parameter ($C$) and the nuclear parameter ($g$) were determined by a grid-searching technique and a leave-one-out cross validation procedure.

RF is an integrated learning algorithm of bagging algorithm and decision tree algorithm, which can fit the complex nonlinear relationship between independent variables and dependent variables (*Wang, Fan & Wang, 2019*). The major parameters in this algorithm were set as follows: the scale was set as "TRUE", and the number of trees (ntree) was set as 500, the number of features tried at each node (mtry) depends on the lowest out-of-bag error. The above three machine learning algorithms were conducted via software R3.5.1 using the packages of "nnet", "e1071" and "randomForest" respectively.

**Table 3 Summary statistics of soil salinity sampling points.**

| Data set | None salinization (<0.2%) | Mild salinization (0.2–0.5%) | Severe salinization (0.5–1.0%) | Min/% | Max/% | CV |
|---|---|---|---|---|---|---|
| Entire dataset ($n = 60$) | 20 | 24 | 16 | 0.08 | 0.81 | 0.54 |
| Calibration ($n = 40$) | 14 | 16 | 10 | 0.08 | 0.81 | 0.54 |
| Validation ($n = 20$) | 6 | 8 | 6 | 0.09 | 0.71 | 0.57 |

The accuracy of the model was evaluated using coefficient of determination ($R^2$), root mean squared error (RMSE) and ratio of performance to deviation (RPD). Model prediction accuracy can be divided into three levels: Level A (RPD > 2.0) indicates very good model prediction; Level B ($1.40 \leq RPD \leq 2.00$) rough quantitative estimation; Level C (RPD < 1.40) unreliable model prediction (*Chang et al., 2001*). A reliable model usually has the characteristics of high $R^2$ and RPD values and low RMSE value. All the three evaluation indicators were calculated via software R3.5.1.

# RESULTS

## Chemical characteristics

With reference to the classification standard for soil salinization degree (*Huang et al., 2018*), the summary statistics are shown in Table 3. The proportion of none salinized soil, mild salinized soil and severe salinized soil in the total samples is 33%, 40%, and 27%, respectively, which is basically consistent with our field investigation. The coefficient of variation (CV) of the total sample was 54%, that is, the variability of soil salinity was not obvious in the cultivated area of the irrigation area.

The statistics of SSC for the entire dataset, calibration dataset and validation dataset are shown in Fig. 5. The statistical indicators of the entire dataset were close to that of the calibration dataset and the validation dataset, indicating that the SSC of the two selected datasets represented the entire dataset. Compared with the field survey results, the degree of the salinization at the sampling points can truly represent that of the cultivated land in the irrigated areas.

## The relationship between SSC and spectral indices

Table 4 shows the relationships between the spectral indices and SSC. Correlation analysis showed a significant positive correlation ($p < 0.01$) between SSC and S1, S3, S4, S5, S6-1, BI-1 and NDSI-2, and a significant positive correlation ($p < 0.05$) between SSC and S2, S6-2, SR-2. The correlation coefficients of the measured SSC data and the three 2D indices (RI, DI and NDI) for the two random spectral bands selected in multispectral bands are shown in Fig. 6.

The bands with higher correlation between SSC and RI, DI, NDI, respectively, were mainly concentrated on the red-edge band (B4) and two near infrared bands (B5, B6). Then each 2D index with the optimal band combinations in the three were applied to subsequent variable selection, and their correlation coefficients with SSC were calculated.
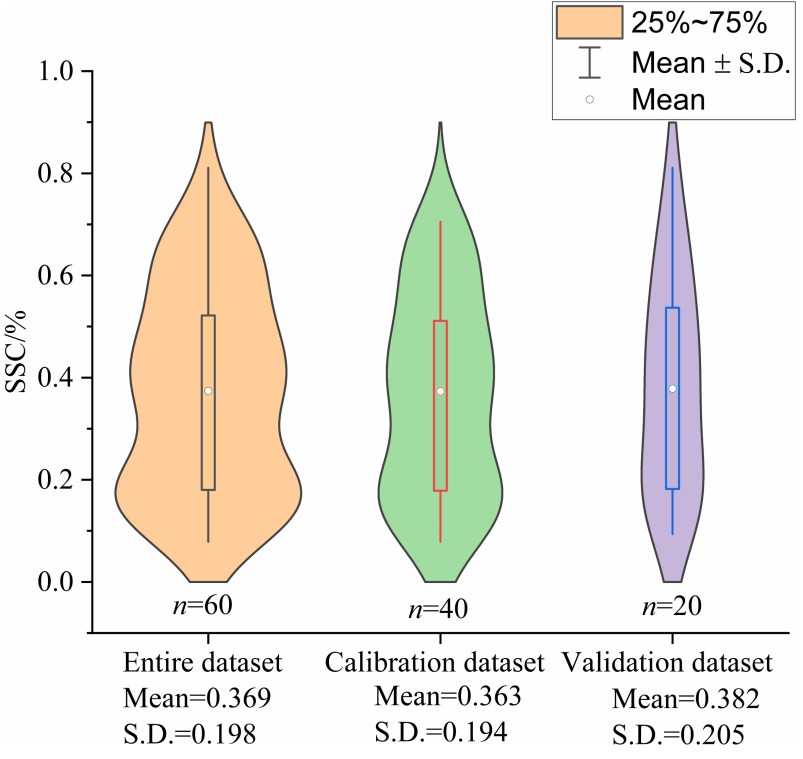

**Figure 5 Violin plots showing the statistics of SSC for entire dataset, calibration dataset and validation dataset (%).** S.D.: standard deviation.

**Table 4 The relationship between SSC and spectral indices.**

| Spectral index | $|R|$ | Spectral index | $|R|$ |
|---|---|---|---|
| S1 | 0.36** | SR-2 | 0.30* |
| S2 | 0.26* | BI-1 | 0.71** |
| S3 | 0.59** | BI-2 | 0.21 |
| S4 | 0.66** | NDSI-1 | 0.19 |
| S5 | 0.43** | NDSI-2 | 0.72** |
| S6-1 | 0.71** | DI | 0.49** |
| S6-2 | 0.27* | RI | 0.59** |
| SR-1 | 0.13 | NDI | 0.55** |

Notes:
* $p < 0.05$.
** $p < 0.01$.

The maximal absolute values of PCC between SSC and RI, DI, NDI, were 0.59, 0.49 and 0.55, respectively.

## Selection of sensitive variables

The GRA of the 22 spectral covariates (six spectral bands, thirteen soil salinity indices and three 2D indices) and measured SSC were conducted by the gray system in the software DPS, and the result is shown in Fig. 7. For the purpose of variable selection, the GCD
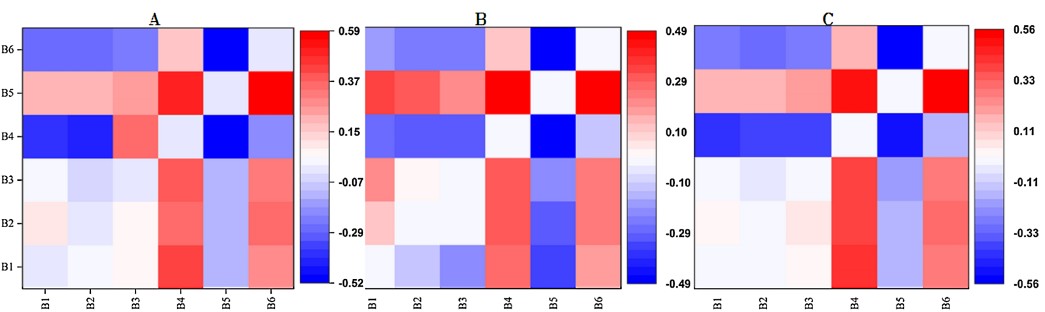

**Figure 6  The correlation coefficients of the measured SSC data and the three 2D indices for the two random spectral bands.** (A) RI; (B) DI; (C) NDI. The color bar on the right side represents the color of Pearson's correlation coefficient (PCC) values. Red stands for positive correlation and blue for negative. The darker the color was, the larger the PCC value was.

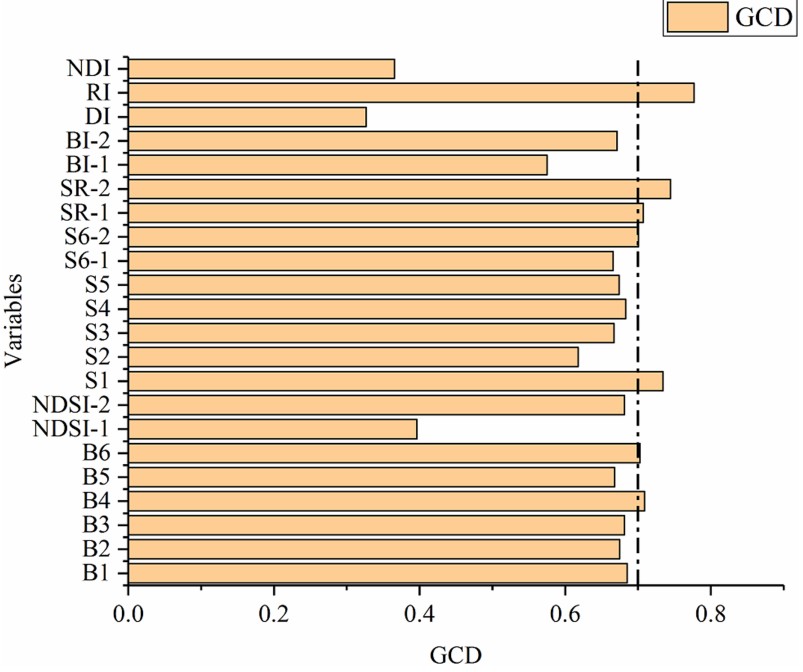

**Figure 7  Gray correlation degree between the variables and SSC.**

threshold of sensitive variables was set as 0.7, and seven variables, including B4, B6, S1, S6-2, SR-1, SR-2 and RI, were finally identified as sensitive variables.

The sensitive variables were selected by SPA method via MATLAB R2014b (Fig. 8) (*Araújo et al., 2001*; *Galvão et al., 2001*). The selected variables were B3 and NDSI, accounting for only 9% of the total number of variables, so the complexity of the model was greatly reduced.

The VIP score of each variable was calculated using MATLAB R2014b (Fig. 9). Then variables with scores higher than 1 were selected, namely, NDSI-2, S1, SR-1, SR-2, BI-1, BI-2, and RI.
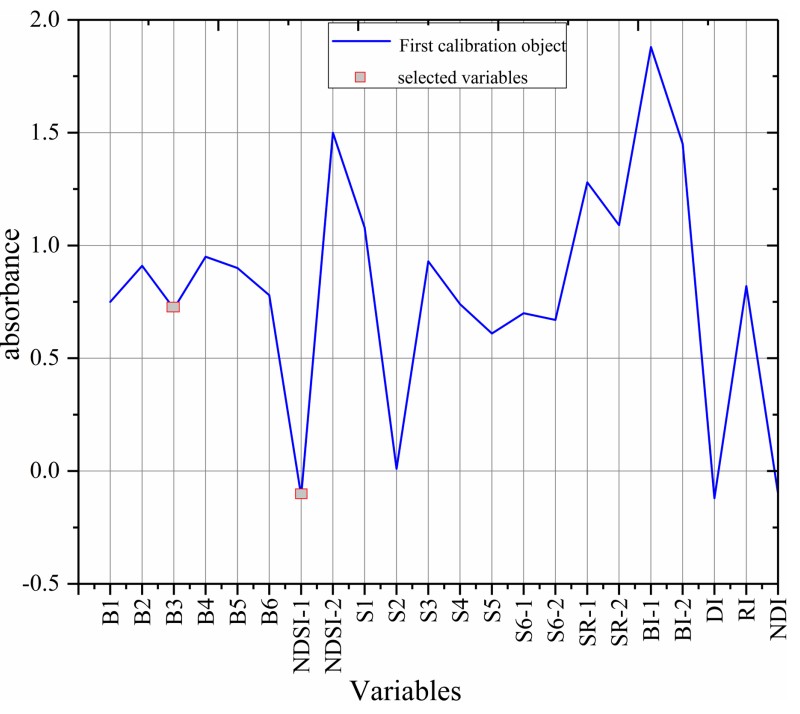

**Figure 8 The selected variables based on SPA.**

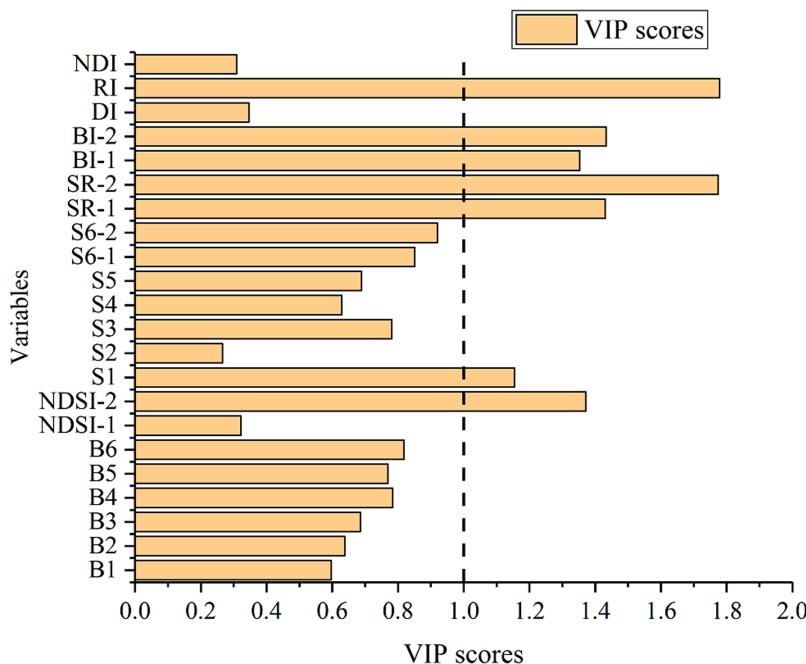

**Figure 9 VIP score of variables for SSC estimation.**

## Model estimations and comparisons

Based on the selection criteria in the previous section, seven sensitive variables (B4, B6, S1, S6-2, SR-1, SR-2 and RI) were selected using GRA; B3 and NDSI-1 were selected using

**Table 5 The details on model parameters.**

| | BPNN | SVR | | RF | |
|---|---|---|---|---|---|
| | size | C | g | ntree | mtry |
| Raw-BPNN | 5 | – | – | – | – |
| GRA-BPNN | 3 | – | – | – | – |
| SPA-BPNN | 2 | – | – | – | – |
| VIP-BPNN | 3 | – | – | – | – |
| Raw-SVR | – | 1000 | 0.01 | – | – |
| GRA-SVR | – | 100 | 0.01 | – | – |
| SPA-SVR | – | 100 | 0.001 | – | – |
| VIP-SVR | – | 1000 | 0.01 | – | – |
| Raw-RF | – | – | – | 500 | 3 |
| GRA-RF | – | – | – | 500 | 3 |
| SPA-RF | – | – | – | 500 | 2 |
| VIP-RF | – | – | – | 500 | 3 |

**Table 6 Comparisons of different machine learning models based on different selection methods.**

| Acronym | $R_c^2$ | $RMSE_C$ | $R_P^2$ | $RMSE_P$ | RPD |
|---|---|---|---|---|---|
| Raw-BPNN | 0.599 | 0.135 | 0.574 | 0.137 | 1.494 |
| GRA-BPNN | 0.661 | 0.116 | 0.677 | 0.116 | 1.764 |
| SPA-BPNN | 0.643 | 0.116 | 0.659 | 0.121 | 1.691 |
| VIP-BPNN | 0.675 | 0.118 | 0.695 | 0.113 | 1.811 |
| Raw-SVR | 0.533 | 0.136 | 0.566 | 0.145 | 1.410 |
| GRA-SVR | 0.645 | 0.120 | 0.625 | 0.131 | 1.562 |
| SPA-SVR | 0.582 | 0.126 | 0.581 | 0.133 | 1.539 |
| VIP-SVR | 0.643 | 0.115 | 0.631 | 0.128 | 1.598 |
| Raw-RF | 0.650 | 0.115 | 0.631 | 0.127 | 1.642 |
| GRA-RF | 0.768 | 0.099 | 0.765 | 0.105 | 1.949 |
| SPA-RF | 0.747 | 0.098 | 0.736 | 0.108 | 1.895 |
| VIP-RF | 0.835 | 0.085 | 0.812 | 0.089 | 2.299 |

**Note:**

Raw, all variables; $R_c^2$, determination coefficient of calibration; $RMSE_C$, root mean squared error of calibration; $R_P^2$, determination coefficient of validation; $RMSE_P$, root mean squared error of validation; RPD, ratio of performance to deviation.

SPA; seven variables (NDSI-2, S1, SR-1, SR-2, BI-1, BI-2, and RI) were selected using VIP. Based on all the variables and the above sensitive variables, three machine learning models (BPNN, SVR and RF) were conducted to estimate SSC. The model parameters are shown in Table 5 and the results are shown in Table 6.

### Analysis of BPNN model

First of all, from the results of the calibration, the $R_c^2$ of the three variable selection methods were close to each other and all bigger than 0.64, and their $RMSE_C$ were all below 0.12. The comparison among the $R_P^2$, $RMSE_P$ and RPD indicated that VIP had the highest

prediction accuracy, GRA came next, and the SPA the lowest. The RPD of the three selection models were between 1.6 and 1.8, while the estimation performance of Raw-BPNN model showed the lowest values (RPD = 1.494). With reference to the three-level classification of RPD, BPNN models could only roughly estimate SSC quantitatively.

### Analysis of SVR model

In terms of calibration effect, the accuracy of the VIP and GRA were relatively close, and both $R_c^2$ were bigger than 0.64. The SPA did not show strong fitting effect, with the lowest $R_c^2$ (0.581) and highest $RMSE_C$ (0.126). The validation results were similar to the calibration results, and the VIP was only slightly higher than the GRA in prediction accuracy while the SPA had the worst. Raw-SVR model had the worst performance. In summary, the $R_P^2/R_c^2$ of the SVR models based on the three variable selection methods were all close to 1, indicating that the SVR models had good robustness (*Wang et al., 2018b*).

### Analysis of RF model

Raw-RF model had the lowest accuracy among the four RF models. According to the accuracy of comprehensive calibration and validation, all the three selection RF models achieved good effects ($R_c^2 > 0.7$) and the RPD reached up to 1.949, 1.895, and 2.299, respectively. The respective $R_P^2/R_c^2$ of the three selection models were 0.9961, 0.9853 and 0.9725, indicating that there was neither over-fitting nor under-fitting. Therefore, it can be concluded that the RF algorithm had excellent robustness and predictive ability.

## Comprehensive evaluation and analysis of the model

Figure 9 shows the performance of the validation models. The estimation accuracy of the models was improved in varying degrees using three variable selection methods, and the values of $R_P^2$ were improved by 0.1 or more. It is interesting that the choice of the different variable selection methods had less difference on model precision. The mean value of RPD of VIP, GRA, and SPA were 1.758, 1.726, and 1.903, respectively. It can be seen from Fig. 10 that the performance of the three selected methods is VIP > GRA > SPA while the mean value of the RPD is not obviously different.

On the premise of the same variable selection method, different machine learning algorithms demonstrated significant differences in the model prediction accuracy. The mean value of the RPD of RF, BPNN and SVR were 2.048, 1.755 and 1.566, respectively. Three RF models displayed high prediction accuracy, while the BPNN models under-fitting and SVR models low prediction accuracy.

Figure 11 shows a comparison between the estimation results of the nine variable selection models and the measured values. The RPD values of all the nine models were bigger than 1.4, suggesting that all models had the ability to quantitatively estimate SSC. The VIP-RF was the model with the best prediction performance (RPD = 2.299), and its validation dataset points were well distributed on both sides of 1:1 line. In addition, the selection of model regression method has a greater impact than that of the variable selection method on the prediction accuracy of SSC.
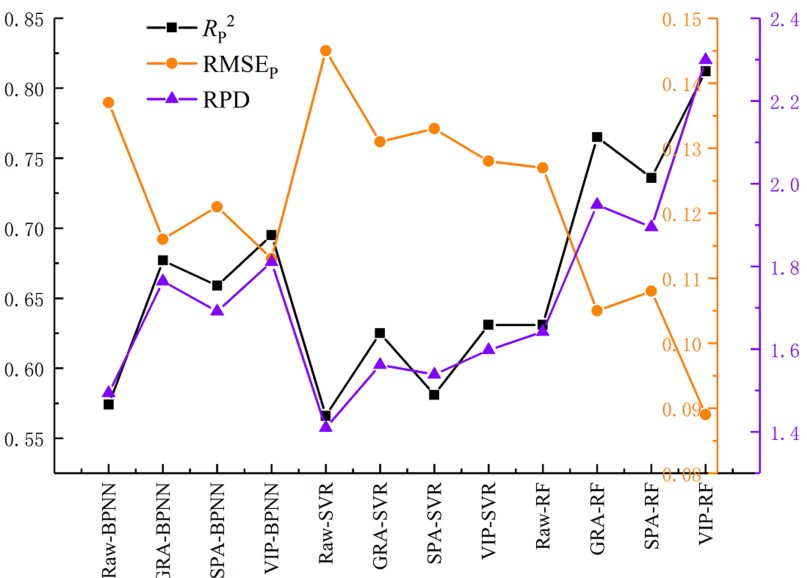

**Figure 10 Prediction performance of the models.**

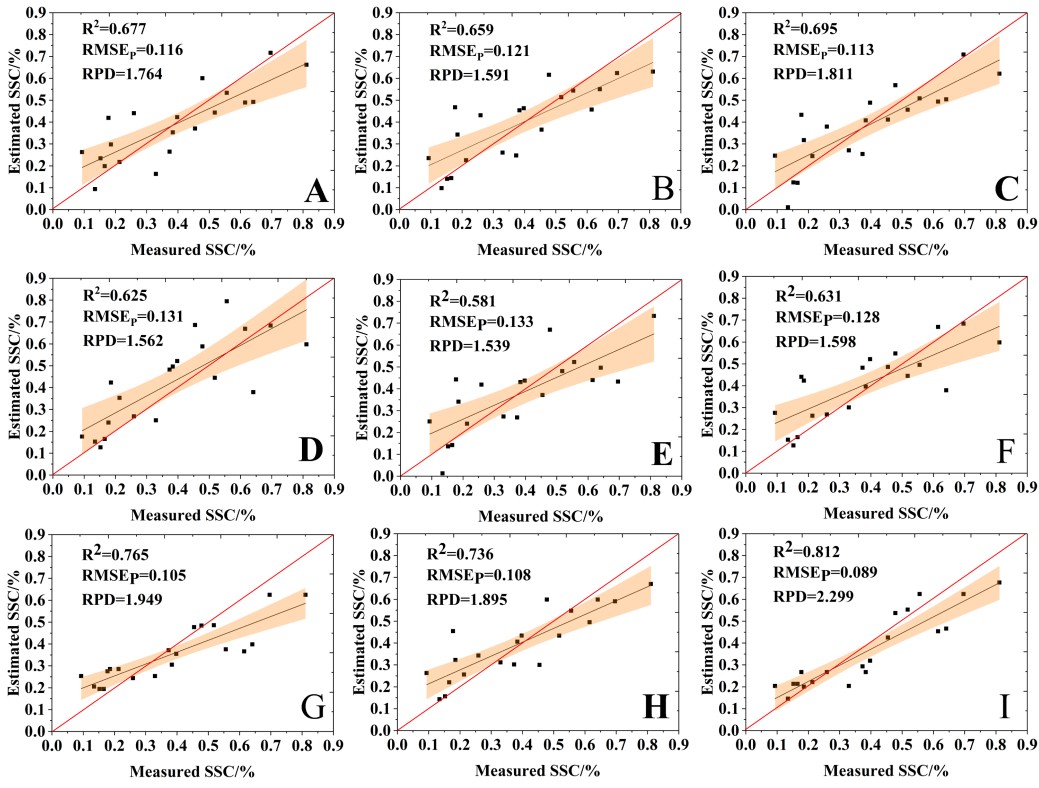

**Figure 11 Comparison of the estimation results of the variable selection models.** (A) GRA-BPNN, (B) SPA-BPNN, (C) VIP-BPNN, (D) GRA-SVR, (E) SPA-SVR, (F) VIP-SVR, (G) GRA-RF, (H) SPA-RF and (I) VIP-RF.

## DISCUSSION

The UAV-borne multispectral RS technique has a great application potential for SSC estimation. Compared with field sampling, UAV cost much lower. Establishing an effective SSC estimation model is of great significance for salinization monitoring in arid and semi-arid areas. Usually, a large number of spectral covariates will be generated in the process of the estimation model construction, but these covariates may contain redundant information, so the selection of sensitive variables is a key step in building an optimal model. In this study, among the twenty-two spectral covariates the ones with higher weights were NDSI, B3 and SR according to the results of the three selection methods. The bands of the three variables were mainly at red band (680 nm) and two near-infrared bands (800 nm, 900 nm), which is consistent with the results of former researches (*Wang et al., 2019b*; *Chen, 2018*). The main water-soluble ions of our study area are $Na^+$, $Ca^{2+}$, $Cl^-$ and $SO_4^{2-}$ (*Wang et al., 2019a*). The visible-near infrared reflectance of salinized soil is higher than that of non-salinized soil (*Weng & Gong, 2006*). *Chen (2018)* proposed that salinized soil had an absorption peak at 671 nm, and *Wang, Fan & Wang (2019)* found that the sensitive band of SSC was at 882–997 nm. These studies revealed that the three bands (680 nm, 800 nm and 900 nm) contained more information related to soil salinity.

In this research, the number of variables selected by the three methods was 7, 7, and 2, respectively, which effectively simplified the estimation model and improved the modeling efficiency. The application of three variable selection methods did improve the accuracy of the estimation models markedly. From the model prediction accuracy, the three selection methods ranked from high to low were: VIP, GRA, and SPA, and this result were consistent with that of *Wang et al. (2019b)*. The SPA method had the lowest accuracy, the possible reason for this result is that the method only selected two variables (9%) and missed some relevant spectral information possibly. However, there was no significant accuracy difference using three different variable selection methods, which may be due to the fact that the number of spectral variables set in this study was not large and these methods were mostly used for hyperspectral wavelength selection. We suppose that the selection of regression model method has a more significant impact on the prediction accuracy than that of variable selection method in that the multispectral estimation model has relatively little spectral information.

All of the estimation models achieved satisfactory results (RPD > 1.4), and $R_p^2/R_c^2$ were close to 1, that is, there was no over-fitting in the prediction results, which indicates that the machine learning algorithm can well fit the complex nonlinear relationship (*Nawar et al., 2016*). The RPD based on the three RF models were 1.949, 1.895 and 2.299, respectively. Different machine learning algorithms had different results, and the RF algorithm demonstrated the best performance. Many scholars drew similar conclusions when comparing regression methods in the estimation models (*Douglas et al., 2018*; *Zeraatpisheh et al., 2019*; *Antoine et al., 2013*). For example, *Gomes et al. (2019)* established the predicting models of soil organic carbon content using RF, SVR, Cubist and Generalized Linear Models (GLM) and the result showed that the RF algorithm achieved

the best prediction result. *Ge et al. (2019)* estimated soil moisture by combining UAV-based hyperspectral imagery and two machine learning algorithms, and concluded the RF models were superior to the ELM models. These researches have demonstrated that RF is an outstanding and stable ensemble-learning algorithm to construct estimation models that can overcome non-linear small sample size, and it has the strong ability to resist overfitting (*Lindner et al., 2015*; *Chen & Liu, 2005*). We think that BPNN and SVR are not necessarily inappropriate algorithms because the prediction results obtained by them are acceptable. SVR belongs to the supervised learning model, and the relative low prediction accuracy of SVR may be due to the fact that SVR models are prone to deviation estimation caused by high noise (*Li et al., 2019*). Besides, SVR model turning can be very tricky and getting the parameters right is difficult. BPNN belongs to neural work model, it is tough to fit, and in our three BPNN models, $R_P^2$ is slightly larger than $R_c^2$, that is, a weak under-fitting phenomenon occurs, which may be the relative small number of verification samples that increased the instability of the prediction results. An effective optimization may occur if the sample size was appropriate.

UAV-borne multispectral RS has the advantages of high resolution and dynamic continuous monitoring. Besides, existing RS data provides a large number of potential resources available, satellite-borne RS data is easy to obtain and covers a large area, portable analytical spectral devices (ASD) provide hundreds of spectral bands. Therefore, we can combine them in the future application, so as to take the particular advantages of multi-source RS data, and form a multi-scale dynamic continuous SSC monitoring network based on the mathematical estimation model.

Few limitations along with questions for future study should be noted in this research. This paper clearly demonstrated that UAV-borne multispectral RS is an effective tool to estimate SSC. It is well known that soil moisture affects the spectral reflectance, and the existence of such difference will be considered in the subsequent studies. The research was conducted during the short bare soil period in May. It is generally considered that it is more meaningful to estimate the SSC during the crop growth period for the development of precision agriculture. Further researches can focus on establishing SSC model via spectral information of the crop canopy. In addition, estimating SSC by combining machine learning algorithms with RS at different scales will be a future direction.

## CONCLUSION

In this study, we built twelve estimation models of SSC on the basis of UAV-borne multispectral data during the bare soil period in the cultivated area. We finally came to the following conclusions.

Firstly, the RPD values of all models were greater than 1.4, indicating all models have an ability to quantitatively estimate SSC, so UAV-borne multispectral remote sensing is feasible for quantitative SSC estimation.

Secondly, the performance of the models has been improved markedly using the three variable selection methods, and the accuracy varied among the three methods: VIP > GRA > SPA, but this difference was not significant.

Thirdly, the choice of different machine learning algorithms had a great effect on the prediction accuracy of the model. In general, the RF had the highest prediction accuracy and strongest robustness, the SVR followed, and the BPNN had the lowest. The VIP-RF model performed the best among the twelve models with $R_P^2$, RMSE$_P$ and RPD of 0.812, 0.089 and 2.229, respectively.

The UAV-borne multispectral RS has great potential for SSC estimation in the future. This instrument can present an efficient method to decision makers of agriculture and environment management departments.

## ACKNOWLEDGEMENTS

The authors would like to thank the Key Laboratory of Agricultural Soil and Water Engineering in Arid and Semiarid Areas of Ministry of Education and the Institute of Water Saving Agriculture in Arid Areas of China for providing test equipment.
The authors want thank Xintao Wang and Jia Han for their enthusiastic help in the process of writing the paper.

### Funding

The research is supported by the National Key Research and Development Program of China (2017YFC0403302, 2016YFD0200700). Support also came from the Science and Technology Plan Project of YangLing (2018GY-03) and the Fundamental Research Funds for the Central Universities (2452019180). The funders had no role in study design, data collection and analysis, decision to publish, or preparation of the manuscript.

### Grant Disclosures

The following grant information was disclosed by the authors:
National Key Research and Development Program of China: 2017YFC0403302 and 2016YFD0200700.
Science and Technology Plan Project: 2018GY-03.
Fundamental Research Funds for the Central Universities: 2452019180.

### Competing Interests

The authors declare that they have no competing interests.

### Author Contributions

- Guangfei Wei conceived and designed the experiments, performed the experiments, analyzed the data, prepared figures and/or tables, authored or reviewed drafts of the paper, and approved the final draft.
- Yu Li conceived and designed the experiments, authored or reviewed drafts of the paper, and approved the final draft.
- Zhitao Zhang conceived and designed the experiments, authored or reviewed drafts of the paper, and approved the final draft.

- Yinwen Chen analyzed the data, authored or reviewed drafts of the paper, and approved the final draft.
- Junying Chen analyzed the data, authored or reviewed drafts of the paper, and approved the final draft.
- Zhihua Yao conceived and designed the experiments, performed the experiments, authored or reviewed drafts of the paper, and approved the final draft.
- Congcong Lao analyzed the data, prepared figures and/or tables, and approved the final draft.
- Huifang Chen performed the experiments, analyzed the data, prepared figures and/or tables, authored or reviewed drafts of the paper, and approved the final draft.

## Field Study Permissions

The following information was supplied relating to field study approvals (i.e., approving body and any reference numbers):

The Hetao irrigation district administration gave field permit approval (No. 2017YFC0403302).

## Data Availability

The raw measurements are available in the Supplemental Files.

## Supplemental Information

Supplemental information for this article can be found online at http://dx.doi.org/10.7717/peerj.9087#supplemental-information.

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
