# Peer review of "Estimation of soil salt content by combining UAV-borne multispectral sensor and machine learning algorithms"

_PeerJ, doi:10.7717/peerj.9087_

## Round 0.1 · original submission · Major Revisions

The reviewers had many suggestions that will significantly increase the quality of this article. Though you should take all of them into account during your revisions, I think the comments made by reviewers 2 and 4 will be the most helpful in strengthening the manuscript. I draw your attention in particular to Items 1-5 in reviewer 2's comments, and Reviewer 4's comments about explaining the underlying chemistry that allows this approach to work in the experimental design as well as matching the strength of conclusions to the discussion of the same conclusions. I also recommend that you review the manuscript for professional English presentation, to remove colloquial language and improve the overall presentation.

Reviewer 1 ·

Basic reporting

The presentation of this work is well down, especially Abstract part is well written. But Some minor problem need attention.
Line 37: It is recommended not to appear “first”, “Subsequently”, and “finally” in the method description section.
Line 46: The result description keeps the tense consistent and should use the past tense. Such as: “the result showed” corresponding is “had been improved”.
Line 61-64: “the process of salinization……”, this sentence is not native. It is recommended to rewrite it in two sentences.
Line 64 and line 199: “To manage” instead of “in order to manage”, “to find” instead of “in order to find”
Line 81-85: This explanation is taking forever. It is recommended to revise.
Moreover, the authors should pay attention to the uniform format of the article paragraph.

Experimental design

Sample selection and experimental design are satisfactory. I would like the author to explain why not consider RPIQ as a model validation indicator. Because SSC is generally non-normal. “software Pix4D” should include information such as version and company.

Validity of the findings

1. Data is robust, statistically sound, & controlled.
2. Conclusions are well stated, linked to original research question & limited to supporting results.
3. Speculation is welcome.

Additional comments

1. “variable selection method” is a better way of putting it than so-called “dimensionality reduction”. VIP, SPA and methods are mainly for variable optimization.
2. Figure 6 is not clear, and the value of the diagonal should be NaN. (B) omit Y-axis.
3. I have not found the relevance of the previously mentioned published indices and SSC.
4. Figure 10, red Y-axis should put near the blue Y-axis.
5. Three machine learning algorithms belong to three fields respectively, for example, RF belongs to ensemble learning, which advantages it plays (except for being appropriate for small samples).

Reviewer 2 ·

Basic reporting

no

Experimental design

no

Validity of the findings

no

Additional comments

This paper investigates modeling of soil salt content using UAV-borne multispectral data. Machine learning methods was found to be most useful. The UAV-borne multispectral remote sensing has great potential for estimating SSC in the future; and a suite of optimized spectral indices were proposed to estimate SSC. The topic is very important given the fact that soil salinization is a very important element in sustainable agriculture. This research paper presents an interesting concept toward that front, and interesting to the readers of the journal. I recommend publications with minor revision. summary comments are below:

1. The contents of the manuscript were too much redundant and should be substantially condensed. The advantages of remote sensing in the Introduction section was common knowledge, and should be drastically cut down.
2. It is well known that soil moisture affects the spectral reflectance. The existence of such difference needs to be considered in the subsequent studies, and the reasons for such difference can be analyzed in depth.
3. I noticed a few citations which follow statements that do not accurately reflect the findings of what is being cited (Line 221-222…..). I recommend you review your citations and ensure that the way they are being presented in your paper is accurate.
4. Why the full-covariates models have less good performances? Can we just imagine the less important covariates as some kind of noises, and these noises compromise the models performances?
5. The authors should add some related discussion about the estimation mechanism. I think your paper can best be improved by adding the related contents.
6. Based on what considerations did you determine the threshold values (e.g., 0.7 for GRA and 1.0 for VIP) for band selection methods?
7. Please specify the detailed information of “Pix4D mapper”
8. This paper did not present the relationship between the SSC and other spectral indices. It is better to add this part.
9. Line 423:”CONCLUSION” should be on the next line.
10. Line 424: “nine estimation models” or “twelve estimation models”, please check it.

Reviewer 3 ·

Basic reporting

Line 61, be resulted from or ...is due to...

Line 62, vague, what is the "specific natural conditions"

Line 79, none of these four papers addressed UAV can monitor heavy metals if I am reading the titles of the four references correctly

Line 99, represent soil salinization, not monitor

Line 106, filter the redundant information, not reduce..

Line 141, missing year for Wang et al.

Experimental design

Line 157, uppercase for km2.

Line 158-163, missing references for climate and soil texture

Line 216 and 224, please write out all the variables. And these formulas are complicated, if this is a function in R or other software, readers might not need to see the whole formula since they don't need to calculate it by hand. Probably can put that in the Appendix.

Validity of the findings

In the Result section, authors put a lot of texts describing the tables or the figures. These should be in the table/figure caption. For example: line 285-289, line 297-301.

Line 342, do authors think an improvement of 0.1 for R2 is a real increase?

The sample size is somehow small (60), although I understand a lot of work is needed in the field and lab for these 60 samples. An possible analysis could be performing a kriging based on the 60 samples to map out the whole area, and then you can pick out more than 60 points (say 100 or 200 points) with values calculated from the kriging in GIS, compared with data from UVA monitoring, so you will have more sample size.

Additional comments

This study is trying to examine if using unmanned aerial vehicles can help estimate soil salt content. It compares the soil salt contents estimated from unmanned aerial vehicles-borne multispectral imagery vs. the values from field sampling and laboratory analysis. It also compares three variable selection methods and different machine learning algorithms. Overall, I find this paper is accepted with revisions that I provided above. Mainly is increasing the sample size by performing a kriging to get more points of "true" values for the plot.

Reviewer 4 ·

Basic reporting

The english is not clear and professional and does not meet the journal standards.

Literature and references do provide sufficient background.

Professional figures and tables are present

Self-contained with relevant results to hypotheses.

Experimental design

- Original Primary Research
- Research question is well defined
- Investigation does not sufficiently explain how spectral bands and indices of interest are related to the soil property in question. More fundamental explanation of the underlying chemistry is needed to justify research approach and demonstrate that the models aren't just fitting to noise.
- Additional details on the model parameters is needed in text. There sufficient explanation about features used, but not about the internal model parameters that were selected and why.

Validity of the findings

Train/test split is satisfactory and statistically sound.

Strong conclusions about the performance of the BPNN and SVR models are drawn in the results that are later tempered in the discussion. The strong conclusions in the results need to be removed. See detailed comments for more explanation.

The conclusions are relying on relatively arbitrary model performance criteria to justify success. The performance criteria need to be better justified.

Additional comments

Line 63: be more specific about what you mean by salt damage and remove “salt damage.
Line 100: specify what is meant by satisfying results
Lined 108-110: explain briefly to the reader how VIP and GRA are conducted
Line 168: This sentence is unnecessary
Line 176: Be specific about what treatments were run to remove impurities and what impurities were removed
Line 187: Good field of vision is subjective, be specific
Line 197: Explain why these bands were selected and what soil spectral properties they relate to in order to justify why they were selected.
Line 212: Define what DPS system is
Line 250: SVR is not really new
Line 261: Site the specific R packages that were used.
Line 266-267: These levels are commonly cited in the literature but there isn’t a scientific justification for why they are used. Also Nawar et al 2016 is not the originator of these criteria. Add a scientific justification for why RPD values were used
Line 274: 33, 40 and 27% don’t line up with none, mild and severe.
Line 300: Why was 0.7 selected
Line 323: You can’t conclude this from your study. You do not provide any detail about the BPNN parameters that were used. Neural networks are tough to fit, and you could just have an underfit model because of the specific model parameters used, not because of BPNN architecture generally.
Line 331: Define robustness
Line 352: Shortingcomings for this specific application. Again as with BPNN, SVR model turning can be very tricky and getting the model parameters right is difficult. Be careful in how much you generalize the performance you had to SVR models for this data generally. RF models are the easiest to tune which could be why you are seeing better results.
Line 366: Why were these bands better, what optical or chemical characteristics of the soil related to salinity are related to these features. Explanation is needed.

---

## Round 0.2 · Minor Revisions

Thank you for the updated version of your manuscript. Though it is substantially improved, there are a couple issues that will still need to be addressed.

1) In your updated methods, you state: "The soil samples were received a series of such treatments as...". This needs to be a specific description of methods, stating methods "such as..." is not a methods description. This needs to be improved.

2) L 327 would read better as: "Based on the selection criteria in the previous section, seven sensitive..."

3) The discussion of how spectral response is related to salinity still needs to be improved. in L393 and following, this sentence does not make sense "The visible-near infrared reflectance of salinized soil is higher than that non-salinized soil, and the spectral reflectance of the heavy sodium saline soil is higher than that of the general heavy saline soil". What is the difference between heavy sodium saline and general heavy saline soils? How does that relate to the classifications used in this manuscript? In addition Weng 2006 is not in the references, so there is no way to evaluate this statement.

4) L428: " proved" is not an appropriate word here, "shown" or "demonstrated" would be better

5) Reviewer 3 brings up a point about using AIC. I recognize that AIC is not an appropriate measure for comparing across model types or for RF classifiers, and your use of cross validation is correct inside of each model type. However, if your resultant models are using a different number of predictors, and an adjusted-R2 would be more appropriate than a simple R2

Reviewer 1 ·

Basic reporting

no comment

Experimental design

no comment

Validity of the findings

no comment

Additional comments

This version has been well modified. The idea of using drones to monitor soil salinity is worth promoting. So I recommend to accept and publish.

Reviewer 2 ·

Basic reporting

The authors did an excellent job of responding to my original comments. It is much better than before.

Experimental design

-

Validity of the findings

-

Additional comments

-

Reviewer 3 ·

Basic reporting

None

Experimental design

None

Validity of the findings

None

Additional comments

This paper was well revised. I only have one more question. Line 362, judging which model is better based on R2 is not accurate. Usually R2 will increase with more variables being added into the model. Probably adding AIC although authors did add other two criteria (RMSE, RPD)

---

## Round 0.3 · accepted · Accept

Thank you for addressing the remaining details that had to be completed in this revision round.